# How to talk so AI will learn:
# Instructions, descriptions, and autonomy

**Theodore R. Sumers**
Computer Science
Princeton University
sumers@princeton.edu

**Robert D. Hawkins**
Princeton Neuroscience Institute
Princeton University
rdhawkins@princeton.edu

**Mark K. Ho**
Computer Science
Princeton University
mho@princeton.edu

**Thomas L. Griffiths**
Computer Science, Psychology
Princeton University
tomg@princeton.edu

**Dylan Hadfield-Menell**
EECS, CSAIL
MIT
dhm@csail.mit.edu

## Abstract

From the earliest years of our lives, humans use language to express our beliefs and desires. Being able to talk to artificial agents about our preferences would thus fulfill a central goal of value alignment. Yet today, we lack computational models explaining such language use. To address this challenge, we formalize learning from language in a contextual bandit setting and ask how a human might communicate preferences over behaviors. We study two distinct types of language: *instructions*, which provide information about the desired policy, and *descriptions*, which provide information about the reward function. We show that the agent's degree of autonomy determines which form of language is optimal: instructions are better in low-autonomy settings, but descriptions are better when the agent will need to act independently. We then define a pragmatic listener agent that robustly infers the speaker's reward function by reasoning about *how* the speaker expresses themselves. We validate our models with a behavioral experiment, demonstrating that (1) our speaker model predicts human behavior, and (2) our pragmatic listener successfully recovers humans' reward functions. Finally, we show that this form of social learning can integrate with and reduce regret in traditional reinforcement learning. We hope these insights facilitate a shift from developing agents that *obey* language to agents that *learn* from it.

## 1   Introduction

As artificial agents proliferate in society, aligning them with human values is increasingly important [1–3]. But how can we build machines that understand what we want? Prior work has highlighted the difficulty of specifying our desires via numerical reward functions [3–5]. Here, we explore language as a means to communicate them. While most previous work on language input to AI systems focuses on *instructions* [6–19], we study instructions alongside more abstract, *descriptive* language [20–25]. We examine how humans communicate about rewards and formalize learning from this input.

To consider how humans communicate about reward functions, imagine taking up mushroom foraging. How would you learn the rewards associated with different fungi (i.e. which are delicious and which are deadly)? In such a setting, learning from direct experience [26] is risky; most humans would seek to learn *socially* instead. So how might we learn reward functions from others? Prior work in reinforcement learning (RL) has examined a number of social learning strategies, including passive *inverse reinforcement learning* (observe an expert pick mushrooms, then infer their reward

36th Conference on Neural Information Processing Systems (NeurIPS 2022).

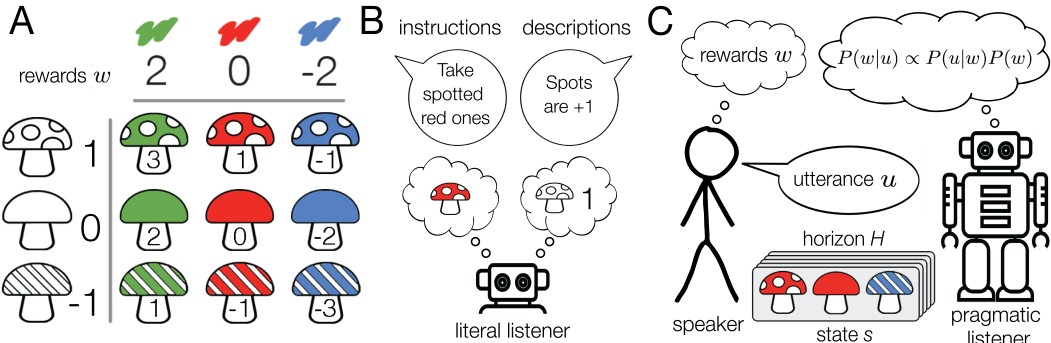

**Figure 1:** We formalize learning from language in a contextual bandit setting. **A**: Linear rewards associated with features determine whether actions (mushrooms) are high or low reward. **B**: We consider two forms of language. *Instructions* tell the listener to take a specific action, while *descriptions* provide information about underlying reward function. A *literal* listener interprets utterances according to these fixed semantics. **C**: We introduce a speaker model suggesting that humans use instructions and descriptions adaptively to maximize a literal listener's expected rewards over some task *horizon*. We assume the start state (i.e., the present) is known to both speaker and listener, but future states are unknown. The horizon determines the total number of states the listener will act in. Short horizons thus formalize low autonomy settings where the agent receives frequent linguistic input, while long horizons are high autonomy. In order to *learn* from linguistic input, we define a pragmatic listener which uses Bayesian inference to recover the speaker's latent reward function.

function [27, 28]) or active *preference learning* (offer an expert pairs of mushrooms, observe which one they eat, and infer their reward function [29–31]).

However, few humans would rely on such indirect data if they had access to a cooperative teacher [32–35]. For example, an expert guiding a foraging trip might *demonstrate* [36, 37] or verbally *instruct* [7] the learner to pick certain mushrooms, licensing stronger inferences. While such pedagogical actions have been useful for guiding RL agents [10, 17–19], natural language affords richer, under-explored forms of teaching. For example, an expert teaching a seminar might instead *describe* how to recognize edible or toxic mushrooms based on their features, thus providing highly generalized information.

To formalize this process, we present a model of learning from language in a contextual bandit setting (Fig 1). We propose a speaker model that chooses utterances to maximize the listener's expected rewards over some task *horizon*. The horizon quantifies notions of autonomy described in previous work [13, 14, 25]. If the horizon is short, the agent is closely supervised; at longer horizons, they are expected to act more independently. We then analyze *instructions* (which provide a partial policy) and *descriptions* (which provide partial information about the reward function). We show that instructions are optimal at short horizons, while descriptions are optimal at longer ones.

Second, we consider how a listener might learn from such a speaker. We define a pragmatic listener that infers the speaker's latent reward function based on their utterances. While prior work suggests pragmatic learning can be vulnerable to model mis-specification [38], we show that jointly inferring the speaker's horizon and reward function can mitigate this risk.

Finally, we conduct an online behavioral experiment which shows our models support strong reward inference. We integrate this social inference with traditional RL and find that it accelerates learning and reduces regret. Overall, our results suggest that descriptive language and pragmatic inference are powerful mechanisms for value alignment and learning.[1]

## 2   Related work

Classic RL assumes that the reward function is given to the agent [26]. However, in practice, it is difficult to specify a reward function to obtain desired behavior [3], motivating *learning* the reward function from social input.[2] Most social learning methods assume the expert is simply acting

---

[1]Code and data are available at `https://github.com/tsumers/how-to-talk`.

[2]Another approach bypasses learning the reward function and allows the human to provide a reward signal directly [39, 40]. This is challenging, however, as people do not typically provide classical reinforcement [41].

optimally [27, 28, 42], but recent *pragmatic* methods [36, 43, 44] instead assume the expert is actively teaching.

**Learning reward functions from observed actions.** When the desired behavior is known (but the reward function is not), inverse reinforcement learning [IRL, 13, 14, 27, 28, 42, 45, 46] can be used to infer an expert's reward function from their actions. However, such approaches face fundamental issues with identifiability: observed behavior can often be explained by multiple reward functions. One solution allows the agent to actively query the human [30, 31, 47–49]. An alternative is to make a stronger assumption: that the human is actively teaching [i.e. behaving *pedagogically*, 36, 43, 44]. We next review work on learning rewards from language, then return to these methods.

**Learning reward functions from instructions.** Instructions use language to communicate specific actions or goals [6–8]. Prior work has used instructions to shape reward functions [15–17] or learn a language-conditioned reward model [9–12]. An alternative approach uses instructions for value alignment. This paradigm assumes instructions reflect underlying preferences and uses them to infer this latent reward function [13, 14]. Our work extends this to incorporate other forms of language: we ask when a human would use an instruction (vs. other forms of language), and—given an instruction (vs. other forms of language)—what an agent should infer about the speaker's reward function.

**Learning reward functions from descriptions.** Rather than expressing specific goals, reward-descriptive language encodes abstract information about preferences or the world. The education literature suggests such rich feedback is crucial for human learning [50, 51]. However, a relatively smaller body of work uses it for RL: by learning reward functions directly from existing bodies of text [20–23] or interactive, free-form language input [24, 25]. Our work provides a formal model of such language in order to compare it with more typically studied instructions. Other related lines of work use language which describes agent *behaviors*. This language, whether externally provided [52–59] or internally generated [60–63], is typically used to structure the task representation or guide exploration. In contrast, we study language describing task-relevant properties of the environment.

**Learning reward functions from pedagogy.** The preceding algorithms all assume that training examples (whether demonstrations or language) are generated by a human that is *indifferent to the learning process*. Recent work has begun to challenge this assumption by considering *pedagogical* settings [34–37]. In particular, the Rational Speech Act framework [64, 65] builds on classic Gricean theory [66] to formulate optimal communication in terms of recursive Bayesian inference and decision-making. These ideas have been applied to a variety of language tasks including reference games [67, 68], captioning [69], and instruction following [70]. Developing analogues of linguistic pragmatics in reinforcement learning — i.e., algorithms that assume data are *intentionally designed to be informative* — is an active area of research [36, 38, 43, 44]. In particular, inverse reward design (IRD, [4]) uses pragmatic inference on numerical reward functions: rather than take the provided reward function literally, IRD quantifies uncertainty over the function to mitigate alignment risk. Similarly, we apply pragmatic inference to reward-related language. Applying IRD principles to reward-related language offers two primary benefits over its non-linguistic formulation. First, language is *accessible* for humans. While traditional IRD is useful when offline training RL agents, inferring rewards from language would be broadly useful for real-world, real-time interactions with non-experts. Second, language can address *future settings*: speakers can refer to actions or features which are not physically present. Speakers can thus provide information about rare or hazardous possibilities (e.g. poisonous mushrooms) which lie outside the listener's experience. We next describe a model for such language use.

# 3   Background

In this work, we consider the problem of *learning from language* in a *contextual bandit* setting. This section introduces models of language use from cognitive science, then describes contextual bandits.

**The Rational Speech Acts framework (RSA).** Our theoretical approach is based on the Rational Speech Acts framework (RSA, [64, 65, 71–73]), a computational model of language understanding. RSA uses theory-of-mind reasoning to explain how human listeners derive meaning from words. It begins by defining a *literal listener* $L_0$ which considers only the conventional (e.g., dictionary) definitions of words. These conventions are represented by a "lexicon" function $\mathcal{L}$ which maps

from utterances to world states $w$.[3] RSA then assumes that a *speaker* $S_1$ makes a noisy-optimal selection from a finite set of utterances, choosing $u$ according to some utility function $U(\cdot)$. Finally, a *pragmatic listener* $L_1$ infers the intended meaning by inverting the speaker model:

$$P_{L_0}(w \mid u) \propto \mathcal{L}(u, w) \tag{1}$$

$$P_{S_1}(u \mid w) \propto \exp\{\beta_{S_1} \cdot U(u, w)\} \tag{2}$$

$$P_{L_1}(w \mid u) \propto P_{S_1}(u \mid w)P(w) \tag{3}$$

In Eq. 2, $\beta_{S_1} > 0$ is an inverse temperature parameter. RSA models typically assume the speaker's objective is *informativeness*. Formally, the speaker tries to reduce the listener's information-theoretic uncertainty [78] over the true state of the world $w^*$: $U(u, w^*) = \ln P_{L_0}(w^* \mid u)$. Explicitly modeling the speaker as making a rational choice from a set of possible utterances allows the pragmatic listener to enrich the literal meaning of the utterance, inferring additional information about the world.

To take an intuitive example, we consider the classic phenomenon of *scalar implicature* [79–81]. If Alice announces "I ate some of the cookies," a typical listener Bob understands this as "some *but not all*." RSA explains this as follows. First, the word "some" *literally* maps to any world state where Alice ate more than zero cookies (e.g. 1, 2, 3... or all), while "all" maps only to the world state where she ate all of them (Eq. 1). Then, $S_1$ Alice is presumed to choose a maximally-informative utterance (i.e. one that specifies the world state as precisely as possible; Eq. 2). Finally, $L_1$ Bob reasons as follows: if Alice had eaten all of the cookies, saying "I ate *all* of the cookies" would have been more precise and thus preferred. Because she chose *not* to say this, he can *infer* that her statement means "some *but not all*" (Eq. 3). In this work, we define a pragmatic $L_1$ listener which uses RSA to infer the speaker's reward function. Crucially, such inferences are based on theory-of-mind and thus vulnerable to *misspecification* [38]: if $L_1$ Bob's model of $S_1$ Alice is wrong, his inferences will be wrong. We show how a carefully specified speaker model, incorporating both instructions and descriptions, allows our pragmatic listener to robustly infer a speaker's reward function.

**Contextual bandits.** Contextual bandits can be seen as one-step Markov Decision Processes [26, 82], thus isolating the problem of *learning* a reward function from the problem of *planning* sequential actions to maximize it. As such, they are a popular testbed for RL algorithms [83–87]. Formally, we define a set of $A$ possible actions. Actions are associated with a binary feature vector $\phi : A \rightarrow \{0, 1\}^K$ (e.g., a mushroom may be green (or not), or striped (or not)). Following other work in IRL [4, 27, 42], we assume rewards are a linear function of these features (e.g., green mushrooms tend to be tasty):

$$R(a, w) = w^\top \phi(a) \tag{4}$$

so $w$ is a vector that defines the value of each feature (see Fig. 1A). Each task consists of a sequence of $H$ i.i.d. states. We refer to $H$ as the *horizon*. At each time step $t < H$, the agent is presented with a state $s_t$ consisting of a subset of possible actions: $s_t \subseteq A$ (e.g., a patch containing a set of mushrooms). They choose an action $a \in s_t$ according to their policy, $\pi_L : S \rightarrow \Delta(A)$.

## 4 Formalizing learning from language in contextual bandits

While bandits are typically considered as an *individual* learning problem, we instead ask how an agent should learn *socially* from a cooperative, knowledgeable partner. We formalize this by introducing a second agent: a speaker who knows the true rewards $w$ and the initial state $s_0$, and produces an utterance $u$ (Fig. 1C). This utterance may affect the listener's policy; with a slight abuse of notation, we denote this updated policy as $\pi_L(a \mid u, s)$. The listener then uses this policy to choose actions. The horizon $H$ determines how many actions the listener will perform under this updated policy. Intuitively, $H = 1$ represents minimal autonomy (or maximal supervision, i.e. guided foraging), whereas $H \rightarrow \infty$ is maximal autonomy (or minimal supervision, i.e. teaching the listener to forage independently). We first assume $H$ is known to both listener and speaker, then relax this assumption. This framework exposes two interrelated problems. First, what should a helpful speaker say? And second, how should the listener update their policy in light of this information?

---

[3]The lexicon can be seen as the *grounding* of language [74, 75]. Most RSA models, including this work, assume that groundings are mutually known. However, methods have been developed to learn them [24, 25] or allow for uncertainty [76, 77], and we view integrating the problem of grounding as important future work.

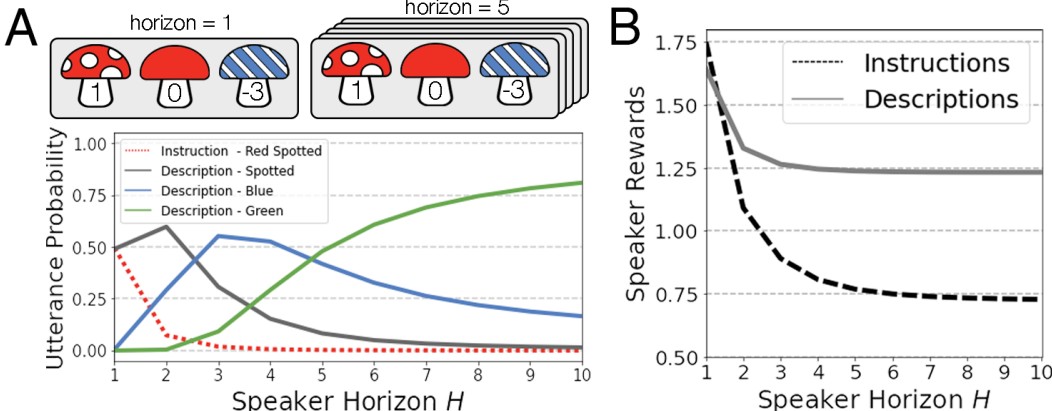

**Figure 2:** The *horizon* formalizes the agent's degree of autonomy and strongly affects a reward-maximizing speaker's choice of utterance. **A**: Top: as the horizon lengthens, the agent will act independently in more unknown future states. Bottom: simulating a reward-maximizing speaker. At short horizons, the speaker is focused on the spotted red mushroom. As the horizon lengthens, they are less biased towards this particular state and provide more generally useful information about the reward function. **B**: Speaker rewards (Eq. 7, averaged over all 84 start states) for a speaker with access to only instructions or only descriptions. Instructions outperform descriptions at short horizons, but descriptions offer much stronger generalization.

## 4.1 Speaker model

We apply the RSA framework (Section 3) to a RL setting by introducing a new speaker objective: instead of Gricean informativeness [65, 66] we assume that speakers seek to *maximize the listener's rewards* [88]. Conceptually, this changes RSA's scope from epistemic to decision-theoretic. Instead of reasoning about how utterances affect the listener's *beliefs*, such a speaker reasons about how they affect the listener's *policy*. When the state is known, we define the *present* utility of an utterance as the expected reward from using the updated policy to choose an action:

$$U_{\text{Present}}(u \mid s, w) = \sum_{a \in s} \pi_L(a \mid u, s) R(a, w) \tag{5}$$

However, we are interested in settings where agents may need to act autonomously [13, 24, 25]. Thus, we must consider how well the policy generalizes to unknown, future states. The *future* utility of an utterance with respect to a distribution over states $P(s)$ can be written as:

$$U_{\text{Future}}(u \mid w) = \sum_{s \in S} U_{\text{Present}}(u \mid s, w) P(s) \tag{6}$$

Because states are i.i.d. in the bandit setting, a speaker optimizing for a horizon $H$ can be defined as a linear combination of Eqs. 5 and 6:

$$U_{S_1}(u \mid w, s, H) = \frac{U_{\text{Present}} + (H - 1)U_{\text{Future}}}{H} = \frac{1}{H}U_{\text{Present}} + (1 - \frac{1}{H})U_{\text{Future}} \tag{7}$$

where $H = 1$ reduces to Eq. 5 and as $H \to \infty$ reduces to Eq. 6. At short horizons, speakers are biased towards producing utterances that optimize the listener's policy in the present state. As the horizon lengthens, the speaker expects the listener to behave more autonomously. Generalization becomes more important, and this bias is reduced.

## 4.2 Formalizing instructions

We now consider how utterances affect the listener's policy. *Instructions* map to specific actions or trajectories [7, 13]; in our work, "instruction" utterances correspond to the nine actions (Fig 1B). Given an instruction, a literal listener executes the corresponding action. If the action is not available, the listener acts randomly:

$$\pi_{L_0}(a \mid u_{\text{instruction}}, s) = \begin{cases} 0 & \text{if } a \notin s \\ \delta_{[\![u]\!](a)} & \text{if } [\![u]\!] \in s \\ \frac{1}{|s|} & \text{otherwise} \end{cases} \tag{8}$$

where $\delta_{[\![u]\!](a)}$ represents the meaning of $u$, evaluating to one when utterance $u$ grounds to $a$ and zero otherwise. An instruction is a *partial policy*: it designates an action to take in a subset of states.

## 4.3 Formalizing descriptions

Rather than mapping to a specific action, descriptions provide information about the reward function [24, 25, 46]. Following [88], we model descriptions as providing the reward of a single feature, similar to feature queries [31]. Descriptions are thus a tuple: a one-hot binary feature vector and a scalar value, $\langle \mathbb{1}_K, \mathbb{R} \rangle$. These are messages like $\langle$Blue, -2$\rangle$. In this work, we consider the set of 6 features $\times$ 5 values in $[-2, -1, 0, 1, 2]$, yielding 30 descriptive utterances. Formally, $L_0$ "rules out" inconsistent hypotheses about reward weights $w$:

$$L_0(w \mid u_{\text{description}}) \propto \delta_{[\![u]\!](w)} P(w) \tag{9}$$

where $\delta_{[\![u]\!](w)}$ represents the meaning of $u$, evaluating to one when $u$ is true of $w$ and zero otherwise. In this work, we assume $P(w)$ is uniform and there is no correlation between weights. The listener then marginalizes over possible reward functions to choose an action:

$$\pi_{L_0}(a \mid u_{\text{description}}, s) \propto \exp\{\sum_w R(a, w) L_0(w \mid u))\} \tag{10}$$

where $\beta_{L_0}$ is again an inverse temperature parameter.[4]

## 4.4 Comparing instructions and descriptions

Prior work suggests that humans use a mix of instructions and descriptions [24, 25]. What modulates this—when should a rational speaker prefer instructions over descriptions? To explore the effects of horizon on utterance utility, we simulate a nearly-optimal speaker ($\beta_{S_1} = 10$). Fig. 1A shows our bandit setting. We assume the listener begins with a uniform prior over reward weights throughout, and use states consisting of three unique actions (giving 84 possible states).

Fig 2A shows how the speaker's choice of utterance varies as the horizon lengthens. At short horizons, the speaker optimizes for present rewards (Eq. 5) and chooses instructions and descriptions that target the "Spotted Red" action. At longer horizons, future rewards (Eq. 6) play a larger role, and the speaker blends the two objectives (Eq. 7) by describing the highly negative blue feature. Finally, at sufficiently long horizons, future rewards dominate and it settles on describing the green feature—which is irrelevant to the start state, but the most important feature for generalization. To quantify how rational speakers should use instructions and descriptions, we repeat the task for all 84 start states using horizons ranging 1-10 and different utterance sets. Fig 2B plots rewards for speakers with access to *only* instructions *or* descriptions, illustrating why this shift from instructions to descriptions occurs. Instructions outperform descriptions at short horizons (achieving the theoretical maximum average reward of 1.75); as the horizon lengthens, however, descriptions generalize better.[5] Overall, as the horizon lengthens, speakers with access to both instructions and descriptions choose descriptions exclusively, producing highly generalizable information (see Appendix A).

## 4.5 Learning from utterances: a pragmatic listener

We now ask how the listener should *learn* from the speaker's utterance. Following RSA (Section 3), a pragmatic listener $L_1$ inverts the speaker model to infer their latent reward function. This approach is closely related to inverse reward design [4]: because the speaker maximizes rewards (Eq. 7), our pragmatic listener effectively treats utterances as *proxy rewards* [89] specified in language.

**Known horizon.** If the speaker's horizon $H$ is known, we can write a $L_1$ listener as:

$$L_1(w \mid s, u, H) \propto S_1(u \mid w, s, H) P(w) \tag{11}$$

Given an instruction, $L_1$ infers the reward weights that would make such an instruction optimal [13, 14]; given a description, $L_1$ can recover information about features that were not mentioned [24, 25]. The $L_1$ listener then chooses actions by substituting this posterior belief into Eq. 10. In practice,

---

[4]Due to the recursive structure of speaker and listener, simultaneously varying $\beta_{S_1}$ and $\beta_{L_0}$ introduces identifiability issues. We therefore fix $\beta_{L_0} = 3$ throughout this work, and only vary $\beta_{S_1}$. This implies that all speakers address the same listener, but do so with different degrees of optimality.

[5]Because our states consist of only three actions, it is often possible to find a description that uniquely identifies the best action. This allows descriptions to perform nearly as well as instructions at $H = 1$. However, as the number of available actions increases, instructions become increasingly advantageous; see Appendix A.

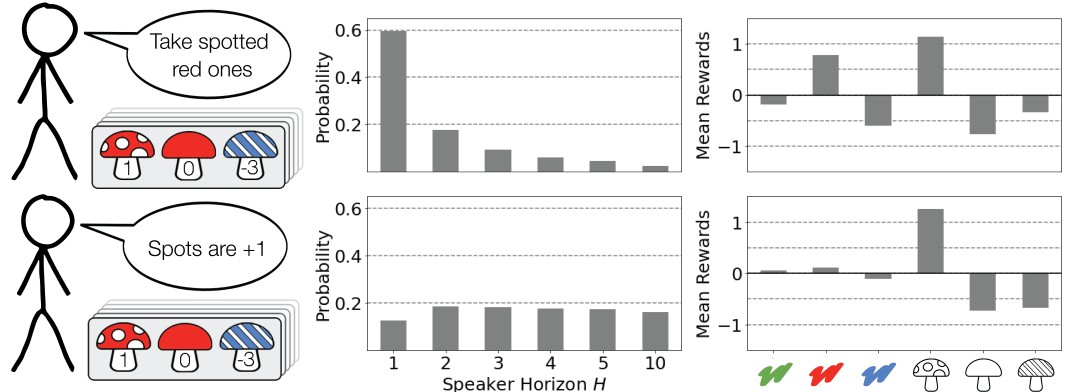

**Figure 3:** Posteriors from latent-horizon pragmatic inference (Eq. 12) for instructions and descriptions. Left: both utterances suggest the listener should take the spotted red mushroom in this state. Center: an instruction (top) suggests the speaker is short-horizon, while a description (bottom) suggests they are long-horizon. Right: this leads to different inference about the speaker's reward function (see Appendix A for full posteriors).

however, the horizon is not known, and so this approach is not feasible for real-world applications. This model instead serves as a theoretically-optimal baseline, and we next consider three practical ways of handling this uncertainty.

**Assuming the horizon.** Perhaps the most straightforward approach is to simply assume a speaker horizon. However, prior work has highlighted the risks of assuming a human is actively trying to teach [38], and suggests that the safest approach is to assume they are not. To test the effects of such mis-specification, in Section 5 we define two pragmatic listener models which assume a short ($H = 1$) or long ($H = 4$) horizon. A pragmatic listener assuming $H = 1$ will *constrain* inference: it will assume the utterance reflects only the speaker's preference between actions in the present state, and thus generalize conservatively. In contrast, a pragmatic listener assuming $H \gg 1$ expects the utterance to generalize broadly, risking overfitting.

**Inferring the horizon.** To mitigate the risk of horizon mis-specification, we can instead assume the speaker's horizon is unknown. Given an utterance, the *latent horizon* pragmatic listener jointly infers both their horizon and rewards, then marginalizes out the horizon:

$$L_1(w \mid s, u) \propto \sum_H S_1(u \mid w, s, H)P(H)P(w) \qquad (12)$$

Intuitively, because short-horizon speakers prefer instructions (and long-horizon speakers prefer descriptions, Fig. 2A), the latent-horizon pragmatic listener can use the utterance type to infer the speaker's horizon and determine the appropriate scope of generalization. To demonstrate this, we simulate a pragmatic listener with a uniform prior over $H \in [1, 2, 3, 4, 5, 10]$. Fig. 3 shows example utterances and resulting inference about the speaker's rewards and latent horizon. Crucially, while both example utterances indicate a preference for the spotted red mushroom, the description suggests the speaker is $H > 1$ and uniquely identifies the spotted feature as high-value.

## 5 Behavioral experiment

To validate our theoretical models, we collected a behavioral dataset. Participants played the role of a mushroom foraging guide and produced utterances for tourists to help them choose good mushrooms. We manipulated the speaker's horizon by varying tourists' itineraries: each tourist was shown visiting a different visible mushroom patch, plus a variable number of unknown future patches (0, 1, or 3, matching horizons 1, 2, and 4 respectively). In the following sections, we analyze the participants' choice of utterances, compare the resulting inference from different listener models, and show how this socially-learned information can accelerate traditional reinforcement learning.[6]

---

[6]This study was approved by the Princeton IRB. The full experiment can be viewed at `https://pragmatic-bandits.herokuapp.com`.

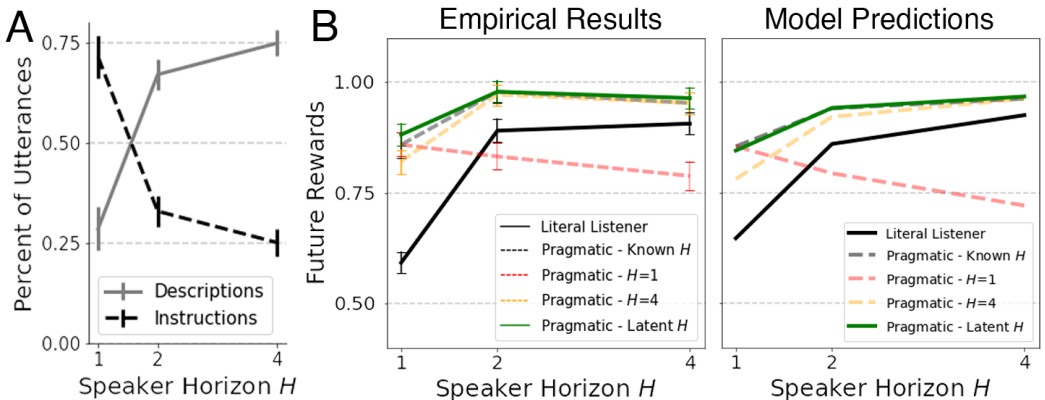

**Figure 4:** Results from behavioral experiment. **A**: Participants preferred instructions when $H = 1$ and descriptions otherwise. **B**, left: empirical future rewards (Eq. 6) from human utterances. As the horizon lengthens, humans choose more generalizable utterances; our latent-horizon listener successfully recovers their reward function. Right: simulations using our speaker model with $\beta_{S_1} = 3$. All error bars show 95% CI.

## 5.1 Experiment setup

We recruited 119 participants on the Prolific experimental platform (`prolific.co`). Participants were trained and tested on the game dynamics then advised a total of 28 tourists. They were told to consider the tourists' itinerary and choose "the most helpful utterance" from drop-down menus allowing them to specify an instruction or description. Because the space of possible descriptions (30) is larger than possible instructions (9), choosing a description required more effort than choosing an instruction. To equalize this, we reduced the set of descriptions by removing neutral features ("red" and "solid") and the 0 value, yielding $4 \times 4 = 16$ possible descriptive utterances. For the remainder of the paper, all pragmatic listeners assume the speaker chooses from this reduced utterance set. Participants received no feedback on how their utterances affected tourists' behaviors, ensuring they chose utterances according to their own sense of how to help. After screening out participants who failed comprehension or attention checks, we were left with 99 participants who produced a total of 2772 utterances. For more details on the data collection, see Appendix B.1.

We next use this set of utterances to explore value alignment (whether our models can infer the speakers' reward function) and then integrated social and reinforcement learning (whether this socially-acquired information can reduce regret in traditional RL).

## 5.2 Inferring rewards: value alignment from language

Overall, our results validated our theoretical models. First, participants were sensitive to the horizon manipulation: there was a statistically-significant shift in their utterance choices between horizons (for $H = 1$ to 2, $\chi^2(26, 1856) = 336.1, p < .001$; for $H = 2$ to 4, $\chi^2(26, 1833) = 40.3, p = .04$). Almost all participants (96 out of 99) used a mix of instructions and descriptions, favoring instructions at $H = 1$ and descriptions at $H > 1$ (Fig. 4A). This led to lower literal future rewards when $H = 1$, and higher future rewards at longer horizons (Fig 4B, "Empirical Results"). To calibrate our pragmatic listeners, we tested $\beta_{S_1} \in [1, 10]$ and found that $\beta_{S_1} = 3$ optimized Known $H$ and Latent $H$ listeners (see Appendix B.3 for details). Simulating utterances produced by our speaker model and resulting pragmatic inference shows a close match to theoretical predictions (Fig. 4B, "Model Predictions", see Appendix B.4 for details).

Our pragmatic listeners offered statistically-significant improvements over the literal listener (Table 1). These gains were particularly large when the speaker has a short horizon, matching our simulations. Somewhat surprisingly, our Latent $H$ model ($M = 0.94, SD = 0.39$) outperformed all other models, including the Known $H$ model ($M = .93, SD = .40$), by a significant albeit small margin (mean difference 0.01, paired-samples t-test $t(2771) = 4.18, p < .001$; see Appendix C for other pairwise tests). This is particularly notable because it underscores the inevitability of misspecification [38]: even when we experimentally controlled the horizon, participants did not perfectly follow our theory,

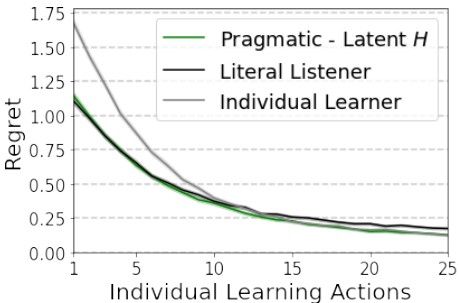

**Figure 5:** Regret with and without social information from our behavioral experiment.

| Listener | Future rewards from utterance | Regret after 25 actions |
|---|---|---|
| Individual | - | 12.14 |
| Literal | .79 | 10.23 |
| Prag - Known $H$ | .93 | **9.42**\* |
| Prag - $H = 1$ | .83 | 9.76 |
| Prag - $H = 4$ | .91 | 9.67 |
| Prag - Latent $H$ | **.94**\*\*\* | 9.55 |

**Table 1:** Results from human utterances. Higher rewards (Section 5.2, Eq. 6) and lower regret (Section 5.3) are better. Bolded results are significantly better.

leading our Latent $H$ model to outperform. We also confirmed the risks of *assuming* a horizon: the two fixed-horizon listeners ($H = 1$ and $H = 4$) underperformed the Latent $H$ listener.

Despite these successes, we found a notable discrepancy between theoretical predictions and empirical results, suggesting a possible future refinement. As described in [88], utility-maximizing speakers agnostic to the truth of an utterance may send messages with exaggerated values (e.g. preferring the false utterance ⟨Spotted, +2⟩ to the true utterance ⟨Spotted, +1⟩). In our experiment, however, participants regularly chose the lower-reward, true utterance (Appendix B). Breaking out results by utterance type indicates that pragmatic gains come primarily on instructions (Appendix D). This suggests that our reward-design objective is in fact too weak: participants' tendency to use true descriptions licenses stronger inference, which we return to in the discussion.

### 5.3   Reducing regret: integrating social and reinforcement learning

We now explore how this form of social learning could augment traditional RL approaches. In particular, we take the inferences from our listeners in the previous section and use them as a prior for reinforcement learning (or in the case of literal instructions, we integrate them into the learner's policy; see Appendix E for details on the experimental setup). We then use Thompson sampling [90–92] to learn the reward function and compare regret over the course of learning for five independent learning trials on each context-utterance pair. We report results for our listener agents, as well as an agent with no social information (the "Individual" agent; Fig. 5, Table 1).

We find that social information can substantially reduce regret. Intriguingly, when comparing regret, our Known $H$ pragmatic listener now achieves the best results; this is likely due to the greater uncertainty inherent in the Latent $H$ inference. To confirm these differences are statistically significant, we use a linear regression to predict regret with a fixed effect of listener and random effects for each of the 2772 utterance-context pairs, again comparing the Latent $H$ listener to all other listeners. This confirms that the Known $H$ listener achieves lower regret than the Latent $H$ listener ($\beta = -.12, t(80400) = -2.14, p = .03$), and all other models suffer higher regret (see Appendix C).

## 6   Discussion

We formalized the challenge of using language to teach an agent to act on our behalf [13, 14, 43, 44, 84] in a bandit setting, introducing the notion of a *horizon* to reflect the agent's degree of autonomy. We find *instructions* are optimal for low-autonomy settings, while *descriptions* are optimal for high autonomy. This distinction allows a pragmatic listener to jointly infer the speaker's horizon and reward function, reducing misspecification risk [38]. Our behavioral experiment supports our theory, demonstrating the benefit of learning from language for value alignment [1–3] and traditional RL [26].

We note several limitations and future directions. First, human participants preferred *truthful* descriptions; this licenses stronger pragmatics [93] such as combining our model with epistemic objectives [64, 65]. Second, we assumed language groundings were known, but future work could incorporate uncertainty [76, 77] or learn them [24, 25, 75]. Third, we studied a simple contextual bandit setting; sequential decision settings will require models of planning, and may consider additional language such as instructions at different levels of abstraction or *transition* descriptions [94]. Lastly,

we modeled a simple interaction consisting of a single utterance. More naturalistic interactions would allow multiple utterances, bidirectional dialogue, or interleaved action and communication [43, 44].

Our work can inform agent design by clarifying how and when different forms of linguistic input can be useful. Instruction following may be optimal when autonomy is unnecessary or preferences are non-Markovian [95]. However, if autonomy is desired [1–3, 13, 14, 38, 43, 44], agents should be equipped to understand descriptions of the world or our preferences [21–25]. Finally, pragmatic agents can *infer* whether preferences are local or general. This suggests that learning from a wide range of language is a promising approach for both value alignment and RL more broadly.

## Acknowledgments and Disclosure of Funding

We thank Rachit Dubey, Karthik Narasimhan, and Carlos Correa for helpful discussions. TRS is supported by the NDSEG Fellowship Program and RDH is supported by the NSF (grant #1911835). This work was additionally supported by a John Templeton Foundation grant to TLG (#61454) and a grant from the Hirji Wigglesworth Family Foundation to DHM.

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
