# OpenReview forum: "How to talk so AI will learn: Instructions, descriptions, and autonomy"
_NeurIPS.cc/2022/Conference — NeurIPS 2022 Accept_

### Official Review · Reviewer_feLE · 2022-07-07

**Rating:** 6
**Confidence:** 2
**Soundness:** 2 fair
**Presentation:** 3 good
**Contribution:** 2 fair

**Summary:**

The goal of this paper is considering a social learning setup to see how humans might communicate preferences over behaviours. The motivation for this is value alignment; making models understand what we want. It's difficult to specify desired in numerical reward functions, so this paper explores language as a means of communicating preferences.

The running example environment is foraging for mushrooms. This is formalised through linear bandits with a listener agent that takes actions (choosing a mushroom) and gets a reward based on a weighted linear combination of the features that make up the mushroom (e.g. spotted green). The listener agent takes actions at each time step in the horizon H. Each feature has a weight, and the combination determines the total reward. Stripes have a reward of -1, but green +2, making a striped green mushroom net-positive.
There's a speaker that produces utterances to maximise the reward of the listener. An utterance can be either an action (e.g. eat the striped green mushroom) or a description (stripes give -1 reward). The speaker produces one utterance per horizon H, so if the horizon is 1 the optimal utterance is to just point to the best action with an instruction, whereas if the horizon > 1, it's more efficient to choose a description that can be re-used at different states of the horizon.

*Literal listener*: interprets utterances literally, executes instructions directly (if utterance is "striped green mushroom", it eats that)
*Pragmatic speaker*: (soft-)optimally selects utterances to maximise reward of literal listener, this means that when the horizon is known the speaker just chooses the optimal utterance to maximise reward (instruction or particular description based on the horizon)
*Pragmatic listener*: pragmatic refers to there being some information outside of the literal utterance that influences how the utterance should be interpreted, and this information is unknown. In this setup, the information is the horizon. When the speaker says "take spotted red" or "spotted is +1" both utterances suggest taking the spotted red mushroom, but the former suggests the horizon is 1 and the latter > 1.

The contributions of the paper are presenting a formal model of cooperative social learning in a linear bandit setting, and showing that short-horizon speakers prefer instructions where long-horizon speakers prefer descriptions. The second contribution is showing that jointly inferring the speaker's horizon and reward function can mitigate the risk of model misspecification, and the third contribution is verifying the theoretical model with a behavioural experiment and showing that the model helps in a traditional RL setting.


**Questions:**

- I think what would help me with this paper is reframing it such that the finding is that providing an agent in a linear bandit setup with noisy information about the reward function prior to the choice of action is useful in the early stages of learning where the agent is still very uncertain about the distribution of rewards. (the claim that pragmatic inference helps I have less issues with in its current form), and that this noisy information could potentially be given in the form of language. Then acknowledging all the problems that come with learning directly from descriptions, like grounding, ambiguity, generalising to novel situations, etc, that need to be solved.

- Relatedly, I wonder if relaxing the assumption that grounding is solved would completely change the game. Again, I think the difficult and interesting problem is grounding language to states. I would like to see some discussion on this.

- When reading about the rational speech act and the recursive reasoning in section 3.1, I think an example of how this would work would be helpful and clarifying.

- line 223: "long-horizon speakers prefer instructions" typo: instructions should be descriptions?

**Limitations:**

To some extent, but I'd also like to read a bit of discussion about the limitation of using language to directly attach a reward to an action, as opposed the more general case where getting a reward by following language instructions might take multiple actions.

**Strengths And Weaknesses:**

**Strengths**
- Value alignment by understanding humans through language is an important problem
- Propose the setting of linear bandits with a social aspect (I can't judge whether this is novel due to lack of experience in theoretical RL)
- A behavioural experiment is done to verify the theoretical model and this shows (to some extent) that the theoretical model makes sense.
- The idea that learning from re-usable descriptions can help agents generalise, and these descriptions might be given in language form.

**Weaknesses***
**EDIT** this main weakness is addressed and was not a weakness but a misunderstanding on my part. Rating adjusted.
My main problem with this paper is that I take some issue with the claims about language, and therefore don't directly see the contribution towards learning from descriptions in linguistic form. The way I see it is as follows. What the authors call language in this work is (1) instructions that map 1-1 to actions where the grounding problem is assumed solved (e.g. the instruction "eat the spotted green mushroom" maps to the action "spotted green" and the model knows this a priori) and (2) descriptions that map 1-1 to features "spotted +1" maps directly to spotted mushrooms and their feature weight. These instructions/descriptions also give away the latent information about the horizon of the linear bandit game: if an instruction is used the horizon is short, if a description is used the horizon is longer. I don't think this qualifies as language (i.e. a symbolic reference system using syntax and symbols combined with context into meaning), because there's no notion of context-dependence, there's no notion of syntax, basically these symbols (e.g. instruction "spotted red") are just noisy indices that point directly 1-to-1 to something. In light of this, how I interpret the finding/contribution of this paper is that providing an agent in a linear bandit setup with noisy information about the reward function prior to the choice of action is useful in the early stages of learning where the agent is still very uncertain about the distribution of rewards. Additionally, this information might contain latent information about the potential rewards and jointly inferring this helps. This noisy information might be given in linguistic form, but I don't see how this paper makes a contribution towards actually giving it in linguistic form. I don't directly see how it tells me anything about how "descriptive language and [...] are powerful mechanisms for value alignment and learning" (which is what the authors summarise their result as). I would be happy to be convinced otherwise if the authors could tell me what I'm missing with regards to the formal description of the problem here, as it's certainly possible that I've missed something.
To me, the interesting challenge about learning from descriptions and the example the authors use to introduce their problem, namely learning in a situation where simply exploring through experience is too dangerous (eating the wrong mushroom is dangerous), is how to generalise from language instructions and descriptions (e.g., "don't eat green striped mushrooms but spotted mushrooms are tasty!") is precisely the grounding problem (how does this green striped mushroom relate to the new mushrooms I encounter in the forest?).
That said, the following statements did make me think this paper is about learning rewards directly from language:
- "Yet today, we lack computational models explaining such flexible and abstract language use. To address this challenge, we consider social learning in a linear bandit setting and ask how a human might communicate preferences over behaviors"
"flexible and abstract language use" and "how humans might communicate preferences" are overstated imo
- "Our findings suggest that social learning from a wider range of language—in particular, expanding the field’s present focus on instructions to include learning from descriptions—is a promising approach "
- "Finally, we conduct a behavioral experiment showing our models support strong reward inference from human-chosen utterances."
This makes me think humans will describe situations and the model can learn from those, whereas in reality in the behavioral experiment humans choose from a dropdown menu of the instructions/descriptions, again making it less "utterances" that are part of language and more indices in an English representation.
- the claim that 99 participants "produced a total of 2772 utterances".
Production of utterances make me think the humans in the behavioral study did more than selecting from a dropdown menu.

---

> ### Author Response · Authors · 2022-08-02
> **The Rational Speech Act and language understanding**
>
> Thank you for the time and energy spent reviewing our work! We do think that certain important aspects of our approach were missed, and we will try to clarify these below. In particular, we hope that improving the introduction to the Rational Speech Act framework will clarify both our approach and its relationship to language understanding.
>
> *I don't think this qualifies as language.* Thank you for raising this concern. First, it may be helpful to see our work as addressing a particular subset of natural language understanding. We study how an agent, engaged in a task, should interpret a specific set of task-relevant utterances, assuming an elementary understanding of their semantic meaning.
>
> We recognize that formalizing linguistic communication as a choice from this discrete set may seem unsatisfying, given the full richness of language. However, our language understanding paradigm (the Rational Speech Act framework) is constructed to emphasize the role of theory-of-mind processes by simplifying other aspects of language such as syntax, compositionality, incrementality, grounding, and so on. The interactions between syntax and pragmatics is still an active area of research, but our formulation is straightforward to extend to larger spaces of utterances and more general lexical semantics. For example, the distinction between instructions and descriptions applies similarly to more syntactically complex utterances.
>
> The framework of recursive Bayesian reasoning has proven remarkably effective at capturing human interpretations of phenomena arising in everyday language use, such as metaphorical or nonliteral language. We have updated our introduction of RSA (previewed below) in order to more clearly link our work to this literature and natural language understanding in cognitive science.
>
> > Our theoretical approach is based on the Rational Speech Act framework (RSA), a computational model of language understanding. RSA uses recursive theory-of-mind reasoning to explain how shared context and assumptions about the speaker allow listeners to derive additional meaning from words. It begins by defining a *literal listener* $L_0$ which considers only the conventional (e.g. dictionary) definitions of words. These conventions are represented by a ``lexicon'' function $\\mathcal{L}$ which maps from utterances to world states $w$. [footnote 1] RSA then assumes that a *speaker* $S_1$ makes a noisy-optimal selection from utterances $u$ according to some utility function $U(\\cdot)$. Finally, a *pragmatic listener* $L_1$ infers the intended meaning by inverting the speaker model:
>
> $P_{L_0}(w \\mid u)  \\propto  \\mathcal{L}(u, w)$     (1)
>
> $P_{S_1}(u \\mid w)  \\propto  \\exp{ \\{ \\beta_{S_1} \\cdot U(u,w)} \\}$      (2)
>
> $P_{L_1}(w \\mid u)  \\propto  P_{S_1}(u \\mid w)P(w) $    (3)
>
> > In Eq. 1, $\\beta_{S_1} > 0$ is an inverse  temperature parameter. RSA models typically assume the speaker's objective is *informativeness*. Formally, the speaker tries to reduce the listener's information-theoretic uncertainty over the true state of the world $w^*$: $U(u, w^*) = \\ln P_{L_0}(w^* \\mid u)$. This process allows listeners to make stronger inferences about the speaker's intended meaning by reasoning about alternative utterances: things they could have, but did not, say.
>
> >To take an intuitive example, consider how this model explains the classic phenomenon of *scalar implicature*. If Alice announces ``I ate some of the cookies,'' a $L_1$ pragmatic listener Bob understands this as "some *but not all*." RSA explains this with the recursive reasoning defined above. In Eq. 1, the word "some" maps to any world state where Alice ate more than zero cookies (e.g. 1, 2, 3... or all), while "all" maps only to the world state where she ate all of them. In Eq. 2, $S_1$ Alice  is presumed to choose a maximally-informative utterance (i.e. one that specifies the world state as precisely as possible). Then, in Eq. 3, $L_1$ Bob reasons as follows: if Alice *had* eaten all of the cookies, saying "I ate all of the cookies" would have been more precise and thus preferred. Because she chose not to say this, he then can then *infer* that her statement means  "some *but not all*." In this work, we define a pragmatic $L_1$ listener which uses RSA to infer the speaker's reward function. Crucially, such inferences are based on theory-of-mind and thus vulnerable to *misspecification*: if $L_1$ Bob's model of $S_1$ Alice is wrong, his inferences will be wrong. We show how a carefully specified speaker model, incorporating both instructions and descriptions, allows our pragmatic listener to more robustly infer a speaker's reward function.
>
> >[footnote 1]: The lexicon can be seen as the *grounding* of language. Most RSA models, including this work, assume that groundings are mutually known. However, methods have been developed to learn them or allow for uncertainty, and we view integrating the problem of grounding as important future work.

---

> > ### Author Response · Authors · 2022-08-02
> > **Pragmatics: context-dependence and linguistic forms**
> >
> > We address two specific points raised in the review about language.
> >
> > *There's no notion of context-dependence.* We would like to clarify that this is not accurate. Our pragmatic listener does, in fact, yield context-dependent inferences. For example, the same instruction, given in different states, will result in a different posterior over possible reward functions. To see this, consider the scenario in Figure 2B: given the instruction “Take the spotted red mushroom,” the listener places a higher positive weight on spotted than red (since mentioning spots is uniquely distinguishing, while mentioning red is not). If this instruction were given in a different state, with different objects, this posterior would change. If the blue striped mushroom were exchanged for a blue spotted one, then neither spots nor red would be uniquely distinguishing. Given the same “Take the spotted red mushroom” instruction, the listener would then place an equal positive weight on both spotted and red. We hope that our newly extended introduction to the Rational Speech Act model clarifies this important property.
> >
> > *I don't see how this paper makes a contribution towards actually giving [reward information] in linguistic form.* A central claim of our work is that the specific linguistic form used (instructions vs descriptions) carries important information that a listener can exploit via pragmatic (theory-of-mind) reasoning. We first discuss why we chose to formalize these specific forms, and then review the implications.
> >
> > First, the two forms of language (instructions and descriptions) were chosen based on prior work which allowed people to produce truly natural language. In particular, see the methods and data from Sumers et al 2021 and Lin et al 2022. Our experiment provides further support that this is an important and clear distinction: following our theory, human participants prefer instructions at short horizons and descriptions at long ones (Fig. 3A). This suggests that our model does capture how humans reason about these different forms.
> >
> > Second, our work shows that the linguistic form contains information beyond the literal content of the utterance. Figure 2B is intended to illustrate this. Note that the specific instruction and description shown convey highly similar information about the reward function: both indicate that the spotted red mushroom should be chosen. However, the pragmatic posterior is different from the literal interpretation; and also different between the two forms. This illustrates how Gricean principles of language understanding (Grice, 1975) may be used in reinforcement learning settings to recover information about the speaker’s latent reward function (e.g. value alignment).
> >
> > Sumers, Theodore R., Mark K. Ho, Robert D. Hawkins, Karthik Narasimhan, and Thomas L. Griffiths. "Learning rewards from linguistic feedback." In Proceedings of the AAAI Conference on Artificial Intelligence, vol. 35, no. 7, pp. 6002-6010. 2021.
> >
> > Lin, Jessy, Daniel Fried, Dan Klein, and Anca Dragan. "Inferring Rewards from Language in Context." In Proceedings of the 60th Annual Meeting of the Association for Computational Linguistics (Volume 1: Long Papers), pp. 8546-8560. 2022.
> >
> > Grice, Herbert P. "Logic and conversation." Speech acts. Brill, 1975. 41-58.

---

> > > ### Author Response · Authors · 2022-08-02
> > > **Grounding and sequential decision making**
> > >
> > > *To me, the interesting challenge is grounding. Would relaxing the assumption that grounding is solved completely change the game?* We agree grounding is an interesting challenge, but our aim is to emphasize and address the complex challenges that remain even after grounding is solved. Even if an agent has perfect grounding, they must still infer the intent of the speaker in order to act autonomously in new contexts. Indeed, we explicitly show that having a literal grounding of instructions is not enough to perform effectively in autonomous settings.
> > >
> > > With this said, we agree that there are potentially interesting dynamics that emerge when the literal grounding is relaxed and must be jointly inferred by the agent along with rewards. Prior work in the RSA framework has explored this setting, learning the groundings (Lin et al, 2022) or incorporating uncertainty over them (Degen et al, 2020). For example, Hawkins et al., (2022) showed how joint inference over lexical groundings allows new conventions to form between the teacher and learner. Useful, context-specific meanings gradually become associated with lexical items, enabling more effective communication. These dynamics may open up new strategies for teaching (e.g. if instructions are initially based on less uncertain groundings, but descriptions gradually become more effective as the more abstract features used become learned). Crucially, the core theoretical properties of the ‘fixed semantics’ case we explored, such as the horizon-sensitivity of descriptions relative to instructions, will still be critical to performance in such extended cases. We will update the paper to address this direction for future work.
> > >
> > > Lin, Jessy, Daniel Fried, Dan Klein, and Anca Dragan. "Inferring Rewards from Language in Context." In Proceedings of the 60th Annual Meeting of the Association for Computational Linguistics (Volume 1: Long Papers), pp. 8546-8560. 2022.
> > >
> > > Degen, Judith, Robert D. Hawkins, Caroline Graf, Elisa Kreiss, and Noah D. Goodman. "When redundancy is useful: A Bayesian approach to “overinformative” referring expressions." Psychological Review 127, no. 4 (2020): 591.
> > >
> > > Hawkins, Robert D., Michael Franke, Michael C. Frank, Adele E. Goldberg, Kenny Smith, Thomas L. Griffiths, and Noah D. Goodman. "From partners to populations: A hierarchical Bayesian account of coordination and convention." Psychological Review (2022).
> > >
> > >
> > > *Applications to sequential decision settings.* Thank you– we will expand the discussion of this important extension to our work. Briefly, we believe such settings offer substantial computational challenges and also intriguing new theoretical frontiers. Extensions to full MDPs will require, for example, integrating models of planning. They will also allow for other forms of task-relevant language, such as *transition* descriptions (Rafferty et al., 2011).
> > >
> > > Rafferty, Anna N., et al. "Faster teaching by POMDP planning." International Conference on Artificial Intelligence in Education. Springer, Berlin, Heidelberg, 2011.
> > >
> > > *Correction to line 223:* Thank you! We will fix this.

---

> > > > ### Comment · Reviewer_feLE · 2022-08-05
> > > > **Main weakness addressed, not a weakness but a misunderstanding on my part**
> > > >
> > > > Thanks for the detailed rebuttal, it's clear to me that a lot of careful consideration went into it, and it made me more excited about this work because indeed the authors are correct I had mainly missed the aspect of language that this addresses: pragmatic language can inform underlying reward function. That is an exciting avenue of research. The whole point of the posterior changing based on the different types of language is a good point that I didn't quite take away from reading the paper. I understand how working towards such a question is currently exploratory, i.e. with toy setup in a bandits setting. I additionally agree with the authors that there's no need to make stuff more complex by including the grounding problem when looking at pragmatic understanding. Since my main weakness is adequately addressed, and in fact due to a misunderstanding of parts of the paper on my part, I'll go ahead and change my rating to a 6, "technically solid, moderate-to-high impact paper, with no major concerns with respect to evaluation, resources, reproducibility, ethical considerations".

---

### Official Review · Reviewer_8vXv · 2022-07-10

**Rating:** 6
**Confidence:** 3
**Soundness:** 3 good
**Presentation:** 2 fair
**Contribution:** 2 fair

**Summary:**

The authors present a model of communication between teacher and
student in an action/reward setting (reinforcemnt learning), wherein
the message or communciation contents are either to do explicit
actions (do this or that) or more general descriptions of reward (what
aspects of the world or state tend to lead to positive/negative
rewards). They call it "instruction" (do specific actions) vs
"descriptions" (describing the reward function in more abstract or
general ways, which in the long run might be more effective). The
authors claim that the latter has been less explored in the research.
Through behavioral experiments, in specific limited settings (a linear
bandit setting), the authors show that humans (as teachers) tend to
tailor their communication appropriately, eg depending on the horizon
of the problem: if the human knows the horion of the task is long,
they will pick descriptive messages (but if the horizon say only one
step, they may just utter do this!). The algorithms they develop
("pragmatic agents") together with their experiments and technical
results, provide evidence that this kind of ("social") learning, in
particular learning the teacher/speaker's reward function, can be
achieved and is useful. For instance, the agent that are helped
initially by a human teacher, reach lower regret faster initially,
compared to a pure RL agent without this bias, and those with
descriptive (more general teaching) remain competitive with the pure
RL agent even after many episodes.  The authors conclude that
considering wider means of language use, rather than limiting to just
instructions, has promise for achieving value alignment, between
humans and artificial agents (in RL settings, etc).


**Questions:**

I'll include misc comments, notes, suggestions, and questions here.

In line 48: "We find that short-horizon speakers (focused on a single,
known state) prefer instructions, while long-horizon speakers
(reasoning about future, unknown states) prefer descriptions.", it's
not that speakers are short-horizon, but how the problem is set up for
them (perhaps add a footnote to further clarify this).



footnote 5, 'available' what? in "However, as the number of available
increases,"

Section 3.1: In Pragmatics subsection ..

--> could it be removed? I am not sure if I saw its relevance to the
 rest of the paper (if you keep it, a few issues next).

--> (still in 3.1) Further describe/explain this and subsequent
formulae (this whole subsection is pretty complicated to decipher.. I
am thinking it's better to move it to after the following
subsections..  or remove.. ):

U(u;w^*)=ln P_{L0} (w^* |u)

-> line 103:  What is soft? in "(soft-)optimally"

-> Describe L1 more, say in "of interest emerges at the L1 level."
(a brief concrete example, perhaps?)

-> In fact, 3.2 was more clear and should come first (before 3.1),
especially the helpful statement that describes your particular
problem setting: "While the bandit problem is typically considered an
individual learning problem, we instead ask how an agent should learn
from a cooperative, knowledgeable partner. We formalize this social
learning problem by introducing a second agent: a speaker who knows
the true rewards w and the initial state s0 , and produces an
utterance u." (after 'partner' perhaps add '(speaker).' to make more
clear )

-> [probably a case of notation abuse or misuse] Note on line 122, the
 notation pi_L (a | u, s) is ambiguous (where pi is a policy), since
 the previous paragraph described policies as mapping states to
 action, while here pi_L is being applied to some conditional
 expression (action a given Speaker has uttered utterance u and given
 state s).. I am reading it as a policy that's a function of or is
 modified by, what the speaker has uttered (I think this is your
 intension), but this is ambigous.

-> In line ~132: "Rather than defining the speaker’s utility U (u; w)
 as Gricean informativeness [69] (i.e. inducing true beliefs, Section
 3.1), we suggest that a cooperative speaker should maximize the
 listener’s rewards, thus grounding utility in the listener’s
 subsequent actions.", this is not really a suggesetion, but happens
 to be the way you have set up the problem.  In this setting, indeed a
 cooperative speaker or teacher has to somehow teach the agent (of
 what actions or policy), and it is not surprising or unexpected the
 communication will take such a form (what specific actions to take,
 or more general information about the class of useful actions,
 depending on state or context).

--> I disagree with the subsection (section 3.3) title "Speakers as
 reward designers": speakers (or good teachers), convey (their
 understanding of) the reward function the best they can.. they don't
 'design' rewards (perhaps this title could be understood as the
 speaker trying to shape the reward function that is forming in the
 listener).

--> line 120 (and elsewhere), in 'We formalize this {\em social}
 learning problem', minor comment: to me, this appears to be a very
 specific case of 'social' learning. For instance, social learning,
 can take place by watching what others do (not just what they say),
 and often more than 2 agents are involved (the learner observes and
 learns from more than one 'teacher' and over several
 episodes). Perhaps further clarify, for instance, convert this line
 to: 'We formalize this type of social learning..'

--> [general comment, related to previous one] Why have one (a single)
  utterance only by the speaker? What about multiple rounds of
  interaction? does learning or the analysis become more complicated?
  Some discussion of that would be useful.


**Limitations:**


Yes, I believe so. The authors discuss potential negative societal impact.


**Strengths And Weaknesses:**

Summary of strengths and weaknesses.

Strengths:

-- The novelty of the approach: can broaden the approaches to
teaching/communicating of best ways to behave (value alignment, etc).

-- The behavioral experiments shed some light on how humans, as
teachers or communicators of what to do, can behave. In addition there
are technical and empirical results.

-- Situating the work within the wider context is well done.

Weaknesses:

-- Captures a very limited window to language with a very limited
experimental set up (seems highly special case, or contrived).  Not
clear whether/how this might apply to real human-robot (or robot to
robot) interactions in practice.

-- The results regarding humans (the behavioral experiments) are not
 very suprising: humans do tailor their communication depending on
 context, and the context could be, in part, the horizon of the
 game/task (still, it is useful to show this in actual experiments,
 and agents that can use this to their benefit) (other relevant
 context includes: the level of the listener (which can depend on age,
 education level, ..), what information has already been conveyed, the
 situation both the speaker and listener are in, etc. etc.).

-- The experiments made a number of simplifying assumptions (eg, the
 concreteness of the communication, and as the authors state, ignoring
 the grounding problem).  It is not clear whether they tackle the most
 challenging problems of humon-robot (or robot to robot) interaction.

-- The clarity of the technical results, the formalism, could be
   improved (see below in 'Questions' section).

---

> ### Author Response · Authors · 2022-08-02
> **Responding to weaknesses**
>
> Thank you for taking the time and effort to review our work! We’re glad that you appreciate the novelty of our approach and its applications for value alignment. We very much appreciate your suggestions, and will make a number of changes to address them in the hopes of clarifying how our work builds on models of language use from cognitive science. We will address your comments in order, starting with weaknesses.
>
> **Weaknesses**
>
>
> *Experimental setup is limited / contrived; unclear how it could apply to real interactions.* Thank you for raising this concern! It was shared with another reviewer, so we have shared a similar response with them.
>
> Our setting is motivated directly from prior work on human-robot interactions, but in retrospect these connections were not clearly stated. We summarize them below, and will revise the introduction and discussion to clarify this.
>
> Briefly, our setting and formalisms were inspired by—and intended to inform—a general class of “AI assistants” (Jeon et al 2020, Milli et al 2017, Lin et al 2022 among others). These agents are expected to both follow instructions and learn the individuals’ preferences in order to act more autonomously on their behalf. Our work formalizes this dynamic and extends it to incorporate more rich and varied language. Concretely, the horizon quantifies the agent’s degree of autonomy, while our definition of descriptions adheres closely to the actual human language produced in Lin et al 2022 and Sumers et al 2021. We hope this helps make the setting feel less contrived, and clarifies applications beyond our work.
>
> Jeon, Hong Jun, Smitha Milli, and Anca Dragan. "Reward-rational (implicit) choice: A unifying formalism for reward learning." Advances in Neural Information Processing Systems 33 (2020): 4415-4426.
>
> Milli, Smitha, Dylan Hadfield-Menell, Anca Dragan, and Stuart Russell. "Should robots be obedient?." In Proceedings of the 26th International Joint Conference on Artificial Intelligence, pp. 4754-4760. 2017.
>
> Lin, Jessy, Daniel Fried, Dan Klein, and Anca Dragan. "Inferring Rewards from Language in Context." In Proceedings of the 60th Annual Meeting of the Association for Computational Linguistics (Volume 1: Long Papers), pp. 8546-8560. 2022.
>
> Sumers, Theodore R., Mark K. Ho, Robert D. Hawkins, Karthik Narasimhan, and Thomas L. Griffiths. "Learning rewards from linguistic feedback." In Proceedings of the AAAI Conference on Artificial Intelligence, vol. 35, no. 7, pp. 6002-6010. 2021.
>
> *The behavioral experiment results are not surprising.* As noted, the intent of the behavioral experiment is to (1) evidence the theory (that humans should be sensitive to this particular form of context, in this particular way) and (2) confirm that our model works (that it is important for the agent to model this sensitivity in order to interpret the humans accurately). While such sensitivity may seem unsurprising under our problem formulation, our results suggest that it can explain prior failures of applying pragmatic inference to value alignment settings (e.g. Milli and Dragan, 2020). Finally, we note that “surprising” results in behavioral studies are not necessarily a good thing: they are less likely to replicate! (Open Science Collaboration, 2015)
>
>
> Milli, Smitha, and Anca D. Dragan. "Literal or pedagogic human? Analyzing human model misspecification in objective learning." Uncertainty in artificial intelligence. PMLR, 2020.
>
> Open Science Collaboration. "Estimating the reproducibility of psychological science." Science 349.6251 (2015): aac4716.
>
>
>
> *The experiments made simplifying assumptions.* We agree that our theory and experiments made numerous simplifying assumptions. We have updated the introduction to RSA as well as the discussion to discuss how extensions may relax these assumptions, including e.g. mutual knowledge of grounding.

---

> > ### Author Response · Authors · 2022-08-02
> > **Clarifying pragmatics and its application to our work**
> >
> > *Lack of clarity in introduction, especially regarding pragmatics (section 3.1 / 3.2).* We are sorry that this section was unclear. Pragmatic reasoning is foundational to our approach and we will revise the paper to introduce it more clearly. We will move this subsection, along with the introduction to contextual bandits, to a dedicated “Background” section, and include a brief statement at the top clarifying their relevance to our setting. We preview this section below in the hopes of addressing your questions. Please let us know if this is easier to understand; if not, we are happy to continue revising through the response period. Note that we omit citations here for clarity.
> >
> > > Our theoretical approach is based on the Rational Speech Act framework (RSA), a computational model of language understanding. RSA uses recursive theory-of-mind reasoning to explain how shared context and assumptions about the speaker allow listeners to derive additional meaning from words. It begins by defining a *literal listener* $L_0$ which considers only the conventional (e.g. dictionary) definitions of words. These conventions are represented by a ``lexicon'' function $\\mathcal{L}$ which maps from utterances to world states $w$. [footnote 1] RSA then assumes that a *speaker* $S_1$ makes a noisy-optimal selection from utterances $u$ according to some utility function $U(\\cdot)$. Finally, a *pragmatic listener* $L_1$ infers the intended meaning by inverting the speaker model:
> >
> > $P_{L_0}(w \\mid u)  \\propto  \\mathcal{L}(u, w)$     (1)
> >
> > $P_{S_1}(u \\mid w)  \\propto  \\exp{ \\{ \\beta_{S_1} \\cdot U(u,w)} \\}$      (2)
> >
> > $P_{L_1}(w \\mid u)  \\propto  P_{S_1}(u \\mid w)P(w) $    (3)
> >
> > > In Eq. 2, $\\beta_{S_1} > 0$ is an inverse  temperature parameter. RSA models typically assume the speaker's objective is *informativeness*. Formally, the speaker tries to reduce the listener's information-theoretic uncertainty over the true state of the world $w^*$: $U(u, w^*) = \\ln P_{L_0}(w^* \\mid u)$. This process allows listeners to make stronger inferences about the speaker's intended meaning by reasoning about alternative utterances: things they could have, but did not, say.
> >
> > >To take an intuitive example, consider how this model explains the classic phenomenon of *scalar implicature*. If Alice announces ``I ate some of the cookies,'' a $L_1$ pragmatic listener Bob understands this as "some *but not all*." RSA explains this with the recursive reasoning defined above. In Eq. 1, the word "some" maps to any world state where Alice ate more than zero cookies (e.g. 1, 2, 3... or all), while "all" maps only to the world state where she ate all of them. In Eq. 2, $S_1$ Alice  is presumed to choose a maximally-informative utterance (i.e. one that specifies the world state as precisely as possible). Then, in Eq. 3, $L_1$ Bob reasons as follows: if Alice *had* eaten all of the cookies, saying "I ate all of the cookies" would have been more precise and thus preferred. Because she chose not to say this, he then can then *infer* that her statement means  "some *but not all*." In this work, we define a pragmatic $L_1$ listener which uses RSA to infer the speaker's reward function. Crucially, such inferences are based on theory-of-mind and thus vulnerable to *misspecification*: if $L_1$ Bob's model of $S_1$ Alice is wrong, his inferences will be wrong. We show how a carefully specified speaker model, incorporating both instructions and descriptions, allows our pragmatic listener to more robustly infer a speaker's reward function.
> >
> > >[footnote 1]: The lexicon can be seen as the *grounding* of language. Most RSA models, including this work, assume that groundings are mutually known. However, methods have been developed to learn them or allow for uncertainty, and we view integrating the problem of grounding as important future work.

---

> > > ### Author Response · Authors · 2022-08-02
> > > **Remaining comments / suggestions**
> > >
> > > *Abuse of notation $\\pi_L (a \\mid u, s)$*: your reading is correct; this signifies the listener’s policy conditioned on hearing a particular utterance. We will clarify this and note that it is indeed an abuse of notation.
> > >
> > > *Defining the speaker’s utility as expected rewards is not surprising given the problem setup.* We hope that our revised introduction to the Rational Speech Act framework (above) clarifies why our formulation of utility is an important divergence from prior work. In particular, a large body of published work in psychology and linguistics assumes a speaker’s goal is informativeness, measured as the likelihood of the true hypothesis in the listener’s belief. Our contribution is to provide an alternative measurement of informativeness — namely, the expected reward of the listener’s policy — and to provide evidence for this model with behavioral experiments. Our results suggest that, in decision-theoretic settings involving an autonomous agent, expected rewards are a more suitable model of speaker utility.
> > >
> > > *Subsection title: Speakers as reward designers.* This particular phrasing alluded to the Reward Design (Singh et al, 2009) and Inverse Reward Design (Hadfield-Menell et al, 2017) papers. These works inspired us and we thought that the section title would help readers understand our approach. However, we recognize this section title may seem misleading without that context. You are correct that our goal was to convey that advice and instructions can both be grounded in reward functions for the agent. This, in turn, allows us to model and run inference over a speaker that considers both instructions and advice. To better convey this, we will re-title the section ``Speaker model’’ and add text to clarify the role that reward functions play in our approach.
> > >
> > > Singh, Satinder, Richard L. Lewis, and Andrew G. Barto. "Where do rewards come from." In Proceedings of the annual conference of the cognitive science society, pp. 2601-2606. Cognitive Science Society, 2009.
> > >
> > > Hadfield-Menell, Dylan, Smitha Milli, Pieter Abbeel, Stuart J. Russell, and Anca Dragan. "Inverse reward design." Advances in neural information processing systems 30 (2017).
> > >
> > >
> > > *“Formalize this social learning problem.”* Thank you for this suggested clarification. We absolutely agree social learning takes many forms and our approach formalizes only a narrow subset of the problem. We will update the language to make this clearer throughout the paper. We will re-title Section 3 “Formalizing learning from language in contextual bandits” to clearly delineate the scope of our formality. In general, we chose the term “social learning” in order to contrast with “reinforcement learning”, but we appreciate that this may overstate our work when used imprecisely. If you have any suggestions for a term that would make this distinction without the same concerns, we will be happy to consider them!
> > >
> > > *Why a single utterance / round?* Indeed, we considered a single utterance / round to simplify the analysis and to make a behavioral experiment feasible. In short, the incorporation of multiple utterances makes the speaker’s decision problem much more complex, as the speaker must reason over interactions between current utterance, subsequent listener behavior, and future context/utterance. We consider extensions to multiple utterances or interleaved action / feedback to be important future directions, and will add a note to this extent in the discussion.
> > >
> > > *Corrections / suggestions on line 48, footnote 5, adding (speaker)*: Thank you for these; we will incorporate them.

---

> ### Comment · Reviewer_8vXv · 2022-08-09
> **Acknowledging authors' response**
>
>
> I have read all the reviews and author responses, and I am happy with the author's suggested
> changes.
>
> Thank you.

---

### Official Review · Reviewer_mi6j · 2022-07-10

**Rating:** 7
**Confidence:** 4
**Soundness:** 4 excellent
**Presentation:** 4 excellent
**Contribution:** 3 good

**Summary:**

This paper studies learning reward functions from natural language feedback. It proposes a linear bandit framework to model learning about the reward function by communicating with a teacher through natural language. The paper focuses on the distinction between _instructive_ feedback, i.e., saying which actions to take, and _descriptive_ feedback, i.e., describing which state to achieve. The main thesis is that instrutive feedback is preferred for short planning horizons and descriptive feedback is preferred for long planning horizons. The authors provide theoretical results to support this claim, as well as a behavioral experiment using humans giving and learning from natural language feedback.

**Questions:**

About Sec 3.3.: What's the difference between having the objective of the teacher be to induce true beliefs in the learning agents vs. causing the agent to achieve high utility? Can you elaborate more on choosing the latter rather than the former? Is there some fundamental trade-off between achieving high utility and having true beliefs? I'm guessing that these two goals are highly correlated, and I was surprised by this section which sounds like there is a trade-off.

**Minor suggestions:**

- It could be useful to expand the caption of Figure 1 to be easier to understand without the context of the main paper (for readers skimming the paper)
- Is Figure 1.B based on real humans or the theoretical model? Would be good to include this in the figure caption.
- In most of the paper you talk about "agents", but in the related work section you sometimes refer to "the robot". Are these the same, or are there parts that are specific to robotics? If not, it would be better to use consistent language.

**Limitations:**

The paper could highlight the limitations of choosing a bandit model more clearly. There is a clear trade-off between having a practically applicable model and a model that you can analyze theoretically, which usually motivated bandit models. It would be nice if the paper discussed this more explicitly.

**Strengths And Weaknesses:**

**Strengths**
- The paper studies value alignment, a very important problem for designing robust AI systems. The paper succeeds in making this problem concrete enough for being relevant for practical applications.
- The conceptual insights about different types of language feedback are interesting and novel.
- The authors provide an extensive behavioral experiment with humans providing strong evidence for their hypothesis about instruction and description feedback.
- The proposed theoretical framework seems simple enough for providing meaningful theoretical insights, but general enough to extend to more complex types of interaction in the future. As such the paper could lead to interesting future work.
- The paper is very well-written and easy to understand.


**Weaknesses**
- I find the theoretical results not very surprising or insightful. To me Theorem 1 and to some extend Theorem 2 seem to follow quite directly from the problem setup (I did not check the proofs)
- The notion of planning over a time-horizon H is a bit artificial in the linear bandit setup, as it is not a full RL problem. I think the bandit setup is still interesting, but the paper should make it clearer that this is a repeated 1-step decision problem and there is no real planning involved.

---

> ### Author Response · Authors · 2022-08-02
> **True beliefs vs reward maximizing; motivation for the horizon**
>
> Thank you for taking the time and energy to review our work! We’re glad you found the writing clear and enjoyed the conceptual insights. You ask an excellent question; we will address it first, then briefly respond to suggestions and weaknesses.
>
> *What’s the difference between true beliefs and high utility?* Thank you for asking– this is an important and subtle distinction. We will address it in detail in this response, and (space permitting) update the paper to include more details.
>
> You are correct that the two goals are correlated. However, there is an important conceptual distinction: “belief” can be quantified only with respect to the listener’s knowledge of the reward function, $p_L(w \\mid u)$, whereas expected rewards (“utility”) are quantified with respect to their policy, $\\pi_L(a \\mid u, s)$. There are several resulting implications.
>
> First, it is not clear how to assign a (literal) epistemic utility to instructions. By definition, instructions directly modify the learner’s policy without modifying their beliefs about rewards (intuitively, a purely instruction-following agent does not know why the instruction is optimal). Thus, it is not clear how an epistemic goal of inducing true beliefs could explain the use of instructions, which do not induce beliefs at all. Notably, the line of work in linguistics and cognitive science that RSA emerged from specifically excludes imperative language from its scope for this reason (Roberts, 2012).
>
> Assuming we consider only descriptive utterances (which can be assigned an epistemic value), two important differences remain. First, the goal of maximizing expected rewards from $\\pi_L(a \\mid u, s)$ depends on the state, whereas maximizing true beliefs $p_L(w \\mid u)$ does not. This leads a reward-maximizing speaker to choose different utterances in different states, whereas a purely epistemic speaker would not. In fact, it leads to an entirely different prioritization over utterances.
>
> Finally, false beliefs (such as exaggerations; e.g. saying “Spotted is +2”) can still be high utility. Indeed, our speaker model frequently predicts such utterances, whereas humans appear to restrict themselves to the true utterance “Spotted is +1” (see Figure S12). This discrepancy is discussed in Section 4.2 (lines 274-282) and the discussion (312-315).
>
> Roberts, Craige. "Information structure: Towards an integrated formal theory of pragmatics." Semantics and pragmatics 5 (2012): 6-1.
>
>
> **Suggestions**
>
> *Figure 1 caption.* 1B reflects a theoretical simulated speaker, not human data. Thank you for these suggestions– we will update the caption accordingly.
>
> *Agent vs robot.* Thank you for pointing out this discrepancy. Our work is intended to apply broadly across autonomous agents. We will streamline the language to use the general term “agent” throughout the paper (including the title, where we will use the more general term AI instead).
>
> **Weaknesses / Limitations**
>
> *Theoretical results are not surprising or insightful.* Thank you for sharing this feedback. Other reviewers raised similar concerns. We have decided to remove the proofs in favor of shorter, informal discussions of the conclusions, which we agree are relatively evident. We will use this space to instead expand our introduction of the Rational Speech Act and pragmatics in general, which was severely constrained due to space limitations.
>
> *Limitations of bandits / a horizon H is artificial.* Thank you for raising this issue. We will clarify that we use bandits in order to isolate the problem of learning rewards (our focus) from the problem of planning to optimize them (i.e. sequential decision making).
>
> Additionally, we will clarify why the horizon is introduced: in order to measure autonomy. Briefly, our goal is to develop agents capable of acting autonomously in a human’s interest (without constant and direct linguistic supervision, in the spirit of Milli et al 2017, Jeon et al 2020, Lin et al 2022, *inter alia*). The horizon is intended to quantify the difference between these settings and more traditional instruction following: traditional instruction following is equivalent to $H=1$ in our setup, as the agent is not expected to act autonomously on the human’s behalf in other contexts. As $H$ increases, the agent is expected to act with increasing autonomy in unknown settings. We recognize that this motivation for the horizon was not clear in the original draft and will update the paper to discuss it more explicitly. We will also update the title to “How to talk so your AI will learn: Instructions, descriptions, and autonomy.”

---

> > ### Comment · Reviewer_mi6j · 2022-08-06
> > **Thanks for the response!**
> >
> > Thanks for responding to my questions, this clarifies a few open points. After reading this as well as the other reviews and responses, I will not change my evaluation of the paper, and I think it should be accepted.

---

### Official Review · Reviewer_aJ8p · 2022-07-14

**Rating:** 7
**Confidence:** 4
**Soundness:** 4 excellent
**Presentation:** 4 excellent
**Contribution:** 3 good

**Summary:**

This paper explores the idea of learning-from-teaching in an RL setting, where a teacher can provide either direct instructions or descriptions of a reward function.  The paper focuses on linear bandits, and introduces a sort of theory-of-mind framework whereby a learner can reason about a teacher's mental state. The paper validates their theoretical models with a human experiment, and show close agreement between human performance and the model's predictions.

**Questions:**

None

**Limitations:**

The authors do a good job of addressing limitations and ethical concerns.

**Strengths And Weaknesses:**

I generally liked this paper. The framework is clear, subtle, interesting and potentially useful.

+ I liked the human experiments, and the validation of the model's predictions
+ The mathematical formulation of the model and the setting was clear
+ I appreciated the authors' attention to detail, and the careful buildup of ideas
+ The use of various statistical tests to support conclusions was tidy and well done
+ I especially enjoyed the bits where the agent reasoned explicitly about the speaker's unknown H. I think this sort of explicit reasoning about mental states needs to be more common in RL.
+ Section 4.3 was intriguing, I was glad to see it included

- The paper feels a bit contrived. I found it unsurprising that instructions were better for small H, and descriptions for long H; it almost feels like the problem setting was conceived to support this eventual conclusion. And so while I generally believe the claims in the paper, I have a hard time believing that this will scale much beyond this toy setting.  It always rubs me a /bit/ the wrong way when the story is much bigger than the result.
- Upon reflection, it was unclear to me what generalizable knowledge I should take with me after reading this paper, because if we eliminate the framing, the conclusions seem too closely tied to the specifics of the experimental setup.  Does this really help me in the non-bandit setting? Does this really help me build better teachers? Does this really say anything about building better agents? Does this really tell me when to prefer instructions vs. descriptions?  I appreciate that scientific knowledge must be built brick by brick, and so I am not unhappy with the contribution as it stands, but I do wish that that the ideas felt more generalizable.  (That said, the strengths of this paper outweigh its weaknesses, so I will argue for acceptance.)

---

> ### Author Response · Authors · 2022-08-02
> **Clarifying motivation for problem setting and takeaways**
>
> Thank you for the thoughtful review! We agree that theory-of-mind is underutilized in RL systems and are glad that you found our approach to integrating it clear and interesting. We also believe that the setting is a natural formalism of prior work in the area, and that our approach offers important and generalizable conclusions. We will work to clarify these points in the paper, and discuss specific efforts to do so below.
>
> *The paper feels contrived.* Thank you for raising this concern! It was shared with another reviewer, so we have shared a similar response with them as well.
>
> Our setting is motivated directly from prior formal work on human-robot interactions. In retrospect, however, these connections were not clearly stated. We summarize them below.
>
> Briefly, our problem setting and formalisms were inspired by—and intended to contribute to—a general class of “AI assistants” (Jeon et al 2020, Milli et al 2017, Lin et al 2022 among others). These agents are expected to both follow instructions and learn the individuals’ preferences in order to act more autonomously on their behalf. Our approach is intended to formalize this dynamic. Concretely, for example, the “horizon” quantifies the agent’s degree of autonomy, while our definition of descriptions adheres closely to the actual human language produced in Lin et al 2022 and Sumers et al 2021. We hope these clarifications help make the setting feel less contrived, and clarifies pathways to applications beyond our work.
>
> We will revise the paper to more explicitly discuss the motivation for, and limitations of, our problem setting (e.g. using contextual bandits rather than full sequential decision settings). Finally, we will re-title the work “How to talk so your AI will learn: Instructions, descriptions, and autonomy” to emphasize this connection.
>
> *I found it unsurprising that instructions were better for small H, and descriptions for long H.* We note that introducing an explicit horizon into this setting is a novel contribution which offers substantial theoretical clarity with respect to prior work (for example, Lin et al 2022 use an ad-hoc cost function in their behavioral experiment, and assume all utterances contain a mix of instructive or descriptive content). If the horizon is sufficient to make it obvious when instructions are better than descriptions, we suggest that is a testament to its explanatory power (and the clarity of our formalism) rather than a detriment to our contribution.
>
> *What generalizable knowledge should I take with me?* Thank you for asking this question! In retrospect our discussion failed to clearly articulate this. We have re-written the final paragraph of the paper and preview it below (omitting citations for clarity), then answer remaining questions.
>
> > Our work can inform agent design by clarifying how and when different forms of linguistic input can be useful. Instruction following may be optimal when autonomy is unnecessary or preferences are non-Markovian. However, if autonomy is desired, agents should be capable of understanding descriptions of the world or our preferences. Finally, pragmatic agents can *infer* whether preferences are local or general. This suggests that learning from a wide range of language is a promising approach for both value alignment and RL more broadly.
>
> *Does this really help me in the non-bandit setting?* We certainly hope so! We view this as a very interesting and promising future direction. Indeed, we feel that sequential decision settings would open up other interesting forms of language, such as instructions at varying levels of abstraction or transition-descriptive language (Rafferty et al., 2011).
>
> That said, we acknowledge that doing so will entail numerous challenges, such as integrating models of planning. We have updated the discussion to acknowledge this.
>
> Rafferty, Anna N., et al. "Faster teaching by POMDP planning." International Conference on Artificial Intelligence in Education. Springer, Berlin, Heidelberg, 2011.
>
> *Does this really help me build better teachers?* This is a very interesting question. To be clear, the pragmatic listener is the central focus of our work. Our teacher (speaker) model is primarily intended as a descriptive account of human behavior, to be used as an internal component of this agent. We argue, through our experiments on horizon misalignment, that reasoning about the teacher may be important for designing effective reward learners. Our goal is to inform the design of agents that learn goals/preferences (as described above) and we do not claim that our work will help “build better teachers” per se.

---

### Official Review · Reviewer_vzCm · 2022-07-15

**Rating:** 7
**Confidence:** 5
**Soundness:** 4 excellent
**Presentation:** 4 excellent
**Contribution:** 3 good

**Summary:**

This paper proposes and validates a qualitative distinction between instructions and descriptions as ways of communicating partial information about a reward function.  Briefly, instructions are shown to be more useful over short horizons.  This is framed as a social learning problem, with a pragmatic listener; this resembles previous work on Inverse Reward Design.

This distinction is supported theoretically, and with synthetic and human experiments in a toy linear (contextual) bandit setting (see Figure 1).  The human experiments show strong but not perfect agreement with the theory, and there is a discussion of these results including this discrepancy, which is attributed to humans being more likely to convey literally true information as opposed to the information that would be most useful from a pragmatic point of view.


**Questions:**

Question: Why is “fixed effect of listener and random effects for each of the 2772 utterance-context pairs” used on line 295, but not (If I understand correctly) lines 266-268?


Suggestions:
* Line 62 needs references.
* In line 72, [9-14] should be placed after “infer”
* It should be clarified that this is a *contextual* bandit setting.
* There are typos on 164, 223, 262.
* In Table 1, It is not clear enough what the “future rewards from utterance” column refers to.
* There should be more detail on the selection of $\beta_{S_1}$, to assure the reader that it is not (in some sense) cherry-picked.


**Limitations:**

Yes I believe they have.

**Strengths And Weaknesses:**

Strengths:
* The proposed distinction is insightful and useful, and well described and supported.
* The experiments and analysis are sensible and support the central point of the paper.
* Human experiments are a great addition.
* The experiments on inferring the horizon vs. assuming knowledge of the horizon were interesting and provide an important counter-point to Milli and Dragan [41].



Weaknesses:
* The experimental setting is extremely simple and the work does not demonstrate any immediate practical utility.  A particular limitation with the author's mention is the need to ground natural language instructions and descriptions, which is not addressed in this work.  Overall, however, I think the work still does make a compelling case for further investigation, and the experiments contribute substantially to that case.
* I found the discussion of prior work on instructions misleading, and I think it needs to be more nuanced.  Instructions play a different role in this work versus, e.g., Badhanau et al. [11].  In that work (and other works that I have seen on instruction following), the instructions are meant to be a complete specification of the task/reward, and the method is meant to train a single agent which can be instructed to do different tasks.  In this work, the instruction is meant to be a partial specification of a single task, and the purpose of the method is for the learner/listener to make the best inference possible about the reward from that instruction.
* I found the analysis lines 155-170 of Section 3.4 weak and unconvincing.  This establishes an equivalence between reward design, demonstrations, and instructions, but under fairly strong conditions.  These conditions should be discussed further, As it is unclear how broadly they apply and what happens when they do not apply.  It is also unclear how this analysis is meant to contribute to the central point of the paper.
* The approach is characterized as inverse reward design (IRD), but IRD is a more specific (albeit closely related) problem.

---

> ### Author Response · Authors · 2022-08-02
> **Clarifying statistical tests and addressing suggestions (1/2)**
>
> Thank you for the concise, constructive, and well-informed review! We’re glad you appreciated the distinction between instructions and descriptions, and the resulting insights about various phenomena such as “misspecification” (e.g. Milli & Dragan, 2019). We will first address your questions / suggestions, and then briefly respond to the weaknesses.
>
>
> *Why use different statistical tests to compare models in 4.2 (paired t-tests) vs 4.3 (linear mixed-effects regression)?* Briefly, this is necessary because the structure of the comparison is subtly different. The paired t-tests are simpler and more straightforward to report, but our Thompson sampling procedure (which uses each of the original data points to generate 5 “regret” samples) breaks the pairwise associations. The mixed-effects model accounts for this grouping. It may be helpful to note that a paired t-test is equivalent to a mixed-effects structure model with a random intercept for each pair, so these are the same statistical test.
>
> In more detail:
> * Both analyses begin with 2772 data points (the utterance-context pairs produced in our experiment). Both apply 5 different listener models to each datapoint (the literal listener and four variations on the pragmatic listener). Then:
> * Section 4.2 directly compares each listener model’s output. For each of the utterance-context pairs, each model outputs a single real-valued “future reward” (Eq. 6) based on the listener’s policy, for a total of 2772 data points per model. We then want to compare these distributions. Importantly, these data are “paired”: for each of the 2772 data points, we can directly compare the results for each model. Informally, you can think of the paired t-test as subtracting one model’s “future rewards” from the other, and then using a one-sample t-test to check if the resulting distribution is significantly different from zero.
> * Section 4.3 instead uses the model outputs to initialize 5 Thompson sampling trials. Each trial returns a real-valued regret, for a total of 2772 x 5 = 13860 data points per model. We then want to compare these distributions for each model. Running 5 independent trials per utterance-context pair is important because each trial is noisy. However, because we used each original data point to generate 5 independent “regret” samples, we no longer have a 1:1 correspondence between “regret” samples across models. This means a pairwise t-test is not appropriate. Instead, we use a more complex mixed-effects model. The “random effects” absorb the variance arising from individual utterance-context pairs, leaving the “fixed effects” to capture the difference in models (which is the quantity we want to measure).
>
> *Various formatting, clarity, and reference issues.* Thank you for bringing these to our attention; we will update the paper to correct them.
>
> *Procedure for selecting $\\beta_{S_1}$.* We apologize: we included details in Appendix C.3; but now realize this was not clearly indicated in the main text of the draft. We will update the draft to specifically reference C.3 on line 262 and C.4 on 263. We are also happy to move some of these details into the main text if you feel this is appropriate.

---

> > ### Author Response · Authors · 2022-08-02
> > **Responding to weaknesses (2/2)**
> >
> > *Not addressing grounding.* We agree that grounding is a crucial problem that we did not discuss sufficiently in the paper. We note that prior work in the RSA paradigm has learned these groundings directly (Lin et al, 2022), or incorporated explicit uncertainty over the grounding as a component of the model (Degen et al, 2020; Hawkins et al., 2022). We will discuss these references when we introduce RSA and as possible future extensions to our work.
> >
> > Lin, Jessy, Daniel Fried, Dan Klein, and Anca Dragan. "Inferring Rewards from Language in Context." In Proceedings of the 60th Annual Meeting of the Association for Computational Linguistics (Volume 1: Long Papers), pp. 8546-8560. 2022.
> >
> > Degen, Judith, Robert D. Hawkins, Caroline Graf, Elisa Kreiss, and Noah D. Goodman. "When redundancy is useful: A Bayesian approach to “overinformative” referring expressions." Psychological Review 127, no. 4 (2020): 591.
> >
> > Hawkins, Robert D., Michael Franke, Michael C. Frank, Adele E. Goldberg, Kenny Smith, Thomas L. Griffiths, & Noah D. Goodman (2022). From partners to populations: A hierarchical Bayesian account of coordination and convention. Psychological Review.
> >
> >
> >
> > *Misleading representation of prior work.* We appreciate your point that Bahdanau et al (among others) use instructions to train a supervised reward model for instruction following. We will update the manuscript accordingly and clarify that our approach more closely resembles e.g. Jeon et al, 2020.
> >
> > Jeon, Hong Jun, Smitha Milli, and Anca Dragan. "Reward-rational (implicit) choice: A unifying formalism for reward learning." Advances in Neural Information Processing Systems 33 (2020): 4415-4426.
> >
> >
> > *Section 3.4 analysis (Thm 2) is not contributing meaningfully to the paper.* Thank you for raising this weakness, which is shared with other reviewers. Based on this feedback, we feel that the formal theorems are not substantially improving the clarity of the paper. We will remove both in favor of shorter, informal discussions. We will use this space to expand our introduction of the Rational Speech Act to make the work more accessible.
> >
> >
> > *IRD is a more specific problem.* We agree. We had hoped this characterization would clarify our approach, but other reviewers found it confusing. We will revise the draft to de-emphasize this connection, moving it down into the Pragmatic Listener section (3.7) where it is most directly related.

---

### Meta-Review · Area_Chair_x5o8 · 2022-08-25

**Recommendation:** Accept
**Confidence:** Certain

**Metareview:**

I thank the authors for their submission and active participation in the discussions. The paper investigates language instructions and descriptions as a way to teach a student RL agent. All reviewers unanimously agree that this is a solid paper worthy of acceptance. In particular, reviewers found the paper to be well motivated [vzCm] and well written [mi6j], tacklign an important problem [feLE]. The experiments are convincing [vzCm], and insightful [mi6j], and the method interesting [aJ8p] and novel [8vXv]. Thus, I am recommending acceptance of the paper and encourage the authors to further improve their paper based on the reviewer feedback.

**Award:**

No

---

### Decision · Program_Chairs · 2022-09-14

Accept